 SciPost Phys. Lect. Notes 85 (2024)

# Field theory of collinear and noncollinear magnetic order

**Oleg Tchernyshyov**

William H. Miller III Department of Physics and Astronomy, Johns Hopkins University,
3400 N. Charles St., Baltimore, MD 21218, USA

olegt@jhu.edu

## Abstract

These lecture notes from the 2023 Summer School "Principles and applications of symmetry in magnetism" introduce the reader to the classical field theory of ferromagnets and antiferromagnets.

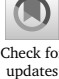

# 1 Introduction

A magnet consists of a large number of atomic magnetic dipoles. The dipoles interact with one another, primarily through the Heisenberg exchange force, which leads to the formation of magnetic order at sufficiently low temperatures. Depending on the sign of the exchange coupling, the exchange energy favors parallel alignment of adjacent magnetic dipoles (ferromagnets), Fig. 1(a), or their antiparallel alignnnment (antiferromagnets). In a simple antiferromagnet, the ordered dipoles alternate between two opposite directions, Fig. 1(b). However, more complex ordered patters are possible when the antiferromagnetic exchange is "frustrated" by the geometry of the atomic lattice and split into 3 or more magnetic sublattices, Fig. 1(c,d).

Atomic magnetic dipoles are associated with the rotational motion of electrons. The vectors of magnetic dipole moment $\boldsymbol{\mu}$ and angular momentum $\mathbf{J}$ are directly proportional to each other:

$$\boldsymbol{\mu} = \gamma \mathbf{J}, \tag{1}$$

where the constant $\gamma$ is the gyromagnetic ratio. The angular momentum of an atom is a

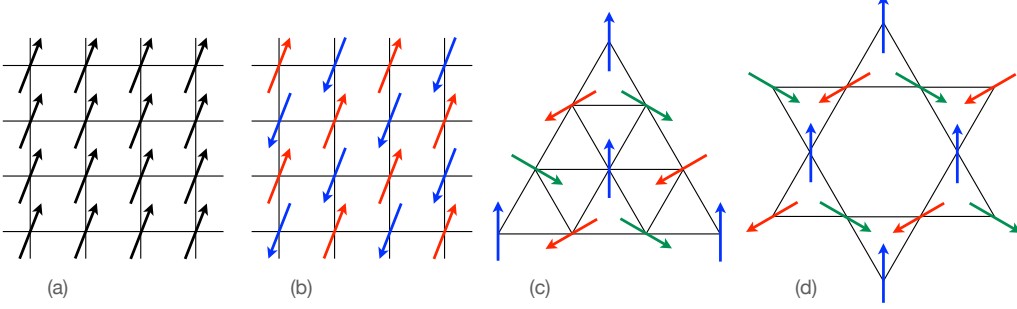

Figure 1: Heisenberg models on various lattices: (a) square, ferromagnet; (b) square, 2-sublattice antiferromagnet; (c) triangular, 3-sublattice antiferromagnet; (d) kagome, 3-sublattice antiferromagnet.

combination of the orbital angular momentum $\mathbf{L}$ and intrinsic spin $\mathbf{S}$ of its electrons. These two types of angular momentum have different gyromagnetic ratios, $\gamma = e/2mc$ for orbital angular momentum and $\gamma = e/mc$ for spin. Here $e = -|e|$ is the electron charge, $m$ is its mass, and $c$ is the speed of light. For simplicity, we will assume that the angular momentum comes from spin only. In view of the direct proportionality (1), the magnetic dipole moment and spin are often used interchangeably.

The spin of an atom $\mathbf{S}$ is a vector of fixed length $S$. In quantum mechanics, $\mathbf{S} \cdot \mathbf{S} = S(S+1)$. Here we use the atomic units, in which $\hbar = 1$. We will deal with the classical dynamics of spin, so $\mathbf{S} \cdot \mathbf{S} = S^2$. It is customary to express the spin in terms of a unit vector $\mathbf{m}$:

$$\mathbf{S} = S\mathbf{m}, \quad |\mathbf{m}| = 1. \tag{2}$$

Here $\mathbf{m}$ refers to the **m**agnetic dipole, which, as we remarked above, is synonymous with spin.

## 2 From one spin to many

### 2.1 Landau–Lifshitz equation for a single spin

The classical dynamics of a spin is similar to that of a fast-spinning top. Its angular momentum $\mathbf{S}$ precesses under the action of an applied torque: $d\mathbf{S}/dt = \boldsymbol{\tau}$. The torque $\boldsymbol{\tau}$ must be orthogonal to the spin vector $\mathbf{S}$, and to its unit-length copy $\mathbf{m}$, to respect the conservation of spin length $S$. Students of magnetism will encounter a great variety of torque types. We will deal with only a few of them here.

The most common torque type, the conservative torque, comes from the dependence of the spin's potential energy $U$ on its orientation:

$$\boldsymbol{\tau} = -\mathbf{S} \times \frac{\partial U}{\partial \mathbf{S}} = -\mathbf{m} \times \frac{\partial U}{\partial \mathbf{m}}. \tag{3}$$

To rationalize this result, think of a spinning gyroscope with a handle attached to its gimbal. By pushing on the handle with a force $\mathbf{F}$, you apply the torque $\boldsymbol{\tau} = \mathbf{r} \times \mathbf{F}$, where $\mathbf{r}$ is the position of the handle. For a conservative force, $\mathbf{F} = -\partial U/\partial \mathbf{r}$, hence $\boldsymbol{\tau} = -\mathbf{r} \times \partial U/\partial \mathbf{r}$, similar to Eq. (3).

The equation of motion for the spin vector $\mathbf{S} = S\mathbf{m}$ under the action of a conservative torque,

$$S\dot{\mathbf{m}} = -\mathbf{m} \times \frac{\partial U}{\partial \mathbf{m}}, \tag{4}$$

is known as the Landau–Lifshitz equation. The dot signifies the time derivative, $\dot{\mathbf{m}} \equiv d\mathbf{m}/dt$.

#### 2.1.1 Precession in a magnetic field

Consider a spin in an external magnetic field $\mathbf{B}$. The spin's potential energy is

$$U = -\boldsymbol{\mu} \cdot \mathbf{B} = -\gamma \mathbf{S} \cdot \mathbf{B}. \tag{5}$$

The Landau–Lifshitz equation (4) then reads $\dot{\mathbf{S}} = \boldsymbol{\Omega} \times \mathbf{S}$, revealing precession at the Larmor frequency $\boldsymbol{\Omega} = -\gamma \mathbf{B}$ about the direction of the magnetic field. This example will give you an idea why the quantity

$$\mathbf{h}_{\text{eff}} = -\frac{\partial U}{\partial \boldsymbol{\mu}} = -\frac{1}{\mu} \frac{\partial U}{\partial \mathbf{m}}, \tag{6}$$

is generally referred to as the effective magnetic field.

## 2.2 Heisenberg spin chain

Consider a chain of identical atoms, each with a spin of the same length $S$. Assume that adjacent spins interact via the Heisenberg exchange of strength $J$. The energy of this system is

$$U = J \sum_n \mathbf{S}_n \cdot \mathbf{S}_{n+1} = JS^2 \sum_n \mathbf{m}_n \cdot \mathbf{m}_{n+1} \,. \tag{7}$$

The nature of the ground states depends on the sign of the exchange constant $J$:

$$\mathbf{m}_n^{(0)} = \begin{cases} \mathbf{m}, & \text{if } J < 0 \text{ (ferromagnet)} \,, \\ (-1)^n \mathbf{n}, & \text{if } J > 0 \text{ (antiferromagnet)} \,. \end{cases} \tag{8}$$

Here $\mathbf{m}$ (for **m**agnetization) and $\mathbf{n}$ (for the **N**éel vector) are arbitrary unit vectors that serve as order parameters in the ferromagnet and antiferromagnet, respectively. They characterize the spontaneous breaking of the global $SO(3)$ symmetry of spin rotations of the Heisenberg exchange energy (7).

It is convenient to measure the energy relative to its ground-state value $U_0 = \pm NJS^2$, where $N$ is the number of bonds in the chain. The upper and lower signs hereafter refer to the ferromagnetic and antiferromagnetic cases, respectively. With the aid of the identity $(\mathbf{m}_{n+1} \mp \mathbf{m}_n)^2/2 = 1 \mp \mathbf{m}_n \cdot \mathbf{m}_{n+1}$, we obtain

$$U = U_0 \mp \frac{JS^2}{2} \sum_n (\mathbf{m}_{n+1} \mp \mathbf{m}_n)^2 \,. \tag{9}$$

The Landau–Lifshitz equation for one spin (4) readily generalizes to the case of many spins:

$$S\dot{\mathbf{m}}_n = -\mathbf{m}_n \times \frac{\partial U}{\partial \mathbf{m}_n} \,. \tag{10}$$

(No summation over $n$ is implied!) Using the Heisenberg exchange energy (9) yields

$$S\dot{\mathbf{m}}_n = -JS^2 \mathbf{m}_n \times (\mathbf{m}_{n-1} \mp 2\mathbf{m}_n + \mathbf{m}_{n+1}) \,. \tag{11}$$

The ground states (8) are stationary states, $\dot{\mathbf{m}}_n = 0$. We will next analyze weak excitations near the ground states, which have the form of spin waves.

### 2.2.1 Spin waves in a ferromagnetic Heisenberg chain.

We examine a weakly excited state $\mathbf{m}_n = \mathbf{m}_n^{(0)} + \delta\mathbf{m}_n$, where $\delta\mathbf{m}_n$ is an infinitesimal deviation from the ground state $\mathbf{m}_n^{(0)}$. Note that $\delta\mathbf{m}_n$ must be orthogonal to $\mathbf{m}_n^{(0)}$ in order to preserve the norm of $\mathbf{m}_n$. Expanding Eq. (11) to the first order in $\delta\mathbf{m}_n$ yields an equation of motion for linear spin waves in a ferromagnet:

$$\delta\dot{\mathbf{m}}_n = -JS\mathbf{m} \times (\delta\mathbf{m}_{n-1} - 2\delta\mathbf{m}_n + \delta\mathbf{m}_{n+1}) \,. \tag{12}$$

Without loss of generality, we choose the ground-state magnetization direction $\mathbf{m} = (0,0,1)$ and express the spin waves as $\delta\mathbf{m}_n = (\text{Re}\,\psi_n, \text{Im}\,\psi_n, 0)$. The complex amplitudes $\psi_n$ satisfy the linear equation

$$i\dot{\psi}_n = JS(\psi_{n-1} - 2\psi_n + \psi_{n+1}) \,. \tag{13}$$

A spin wave $\psi_n(t) = \psi e^{-i\omega t + ikna}$ has the dispersion

$$\omega = 2|J|S(1 - \cos ka) \,. \tag{14}$$

The spin-wave spectrum is shown in the left panel of Fig. 2.

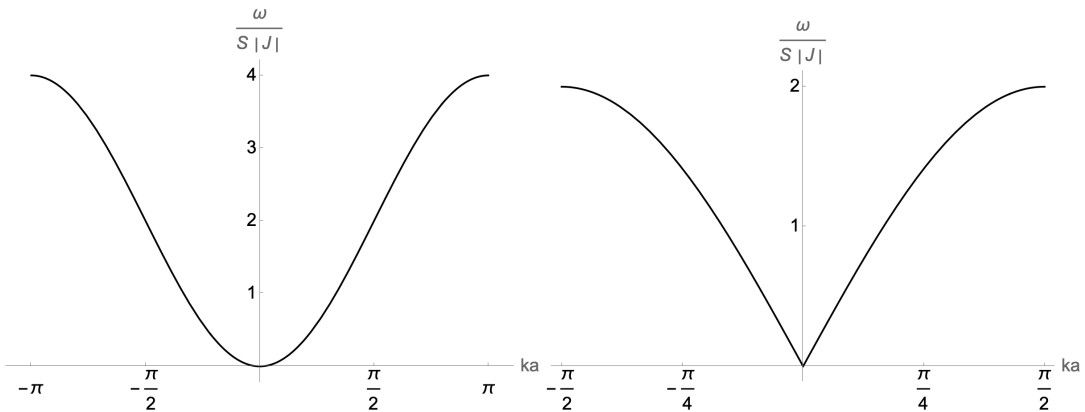

Figure 2: Spin-wave spectra of the classical Heisenberg chain: ferromagnetic (left) and antiferromagnetic (right).

Adding exchange interactions between further neighbors will alter the spin-wave spectrum of the Heisenberg chain. However, some features will remain universal as long as the ground state remains ferromagnetic. One of them is the quadratic dependence of the frequency on the wavenumber, $\omega \sim k^2/2M$ in the long-wavelength limit $k \to 0$, which is reminiscent of a massive nonrelativistic particle with mass $M$. Ferromagnetic spin waves are circularly polarized and precess clockwise—when viewed from the $\mathbf{m}$ direction. As we shall see next, spin waves in the antiferromagnetic chain behave differently.

### 2.2.2 Spin waves in an antiferromagnetic Heisenberg chain.

Expressing the spin variables in terms of the order parameter and spin waves, $\mathbf{m}_n = (-1)^n \mathbf{n} + \delta \mathbf{m}_n$, and expanding Eq. (11) to the first order in $\delta \mathbf{m}_n$ yields the linear spin-wave equation for an antiferromagnet:

$$\delta \dot{\mathbf{m}}_n = (-1)^{n+1} JS\mathbf{n} \times (\delta \mathbf{m}_{n-1} + 2\delta \mathbf{m}_n + \delta \mathbf{m}_{n+1}). \tag{15}$$

The equation of motion looks different for spins with even and odd $n$, reflecting the breaking of translational symmetry by the staggering of the spins in the ground state. The magnetic lattice has a spatial period of $2a$ and contains two atoms per unit (with even and odd $n$).

We set $\mathbf{n} = (0, 0, 1)$ and express the spin-wave part in terms of complex amplitudes separately for the even and odd sublattices:

$$\delta \mathbf{m}_n = \begin{cases} (\operatorname{Re}\psi_n, \operatorname{Im}\psi_n, 0), & \text{if } n \text{ is odd}, \\ (\operatorname{Re}\chi_n, \operatorname{Im}\chi_n, 0), & \text{if } n \text{ is even}. \end{cases} \tag{16}$$

The equations of motion for the complex amplitudes are

$$\begin{aligned} -i\dot{\psi}_n &= JS(\chi_{n-1} + 2\psi_n + \chi_{n+1}), \\ i\dot{\chi}_n &= JS(\psi_{n-1} + 2\chi_n + \psi_{n+1}). \end{aligned} \tag{17}$$

Harmonic waves $\psi_n(t) = \psi e^{-i\omega t + ikna}$, $\chi_n(t) = \chi e^{-i\omega t + ikna}$ have the spectrum

$$\omega = \pm 2JS \sin ka. \tag{18}$$

Positive and negative frequencies correspond to spin waves with clockwise and counterclockwise circular polarization, respectively. The positive part of the spectrum is shown in the right panel of Fig. 2. Note that the Brillouin zone is reduced by half, $|k| \leq \pi/2a$, because of the doubling of the magnetic unit cell.

As we anticipated, spin waves in a Heisenberg antiferromagnet behave differently. The frequency varies linearly with the wavenumber, $\omega \sim sk$ for $k \to 0$. There are now two circular polarizations compared to just one in a ferromagnet.

## 3 Field theory of a ferromagnet

If we are interested in physical properties of our magnets on long length scales—tens of nanometers and longer—field theory offers a more economical description. If we can afford to discard the microscopic details on the scale of the atomic lattice and treat the magnet as a continuous medium, the resulting field theory is much simpler and more versatile. It focuses on universal features of magnets with different underlying atomic lattices and provides valuable connections to the mathematical concepts of symmetry and topology.

### 3.1 From many spins to a spin field

At sufficiently low temperatures, spins in a ferromagnetic chain are nearly parallel to their neighbors, $\mathbf{m}_n \approx \mathbf{m}_{n+1}$. We may thus replace the discrete variables $\mathbf{m}_n$ and their linear combinations with a smoothly varying function $\mathbf{m}(x)$ and its derivatives:

$$
\mathbf{m}_n = \mathbf{m}(x_n),
$$
$$
\mathbf{m}_{n+1} - \mathbf{m}_n = \mathbf{m}(x_n + a) - \mathbf{m}(x_n) = \frac{d\mathbf{m}(x_n)}{dx_n}a + \dots,
$$
$$
\mathbf{m}_{n+1} - 2\mathbf{m}_n + \mathbf{m}_{n-1} = \mathbf{m}(x_n + a) - 2\mathbf{m}(x_n) + \mathbf{m}(x_n - a) = \frac{d^2\mathbf{m}(x_n)}{dx_n^2}a^2 + \dots \tag{19}
$$

### 3.2 Heisenberg ferromagnet

The exchange energy (9) can then be converted from a sum over the atomic site index $n$ to an integral over the continuous coordinate $x$:

$$
U = \frac{|J|S^2}{2}\sum_n [\mathbf{m}(x_n + a) - \mathbf{m}(x_n)]^2 \approx \frac{|J|S^2}{2}\sum_n \left(\frac{d\mathbf{m}(x_n)}{dx_n}\right)^2 a^2 \approx \frac{A}{2}\int dx \left(\frac{d\mathbf{m}(x)}{dx}\right)^2, \tag{20}
$$

where $A = |J|S^2 a$ is the exchange coupling constant in the continuum theory. The energy, previouosly a function of discrete spin variables $\{\mathbf{m}_1, \mathbf{m}_2, \dots, \mathbf{m}_N\}$, is now a functional of the field $\mathbf{m}(x)$. Note that we have shifted the energy by a constant so that a uniform ground state has energy $U = 0$.

The Landau–Lifshitz equation (10) must be adapted for the continuum theory. The partial derivative $\partial U / \partial \mathbf{m}_n$ gets replaced with the functional derivative $\delta U / \delta \mathbf{m}(x)$ and the spin length $S$ with the spin density $\mathcal{S} = S/a$ (in $d = 1$ spatial dimension):

$$
\mathcal{S}\partial_t \mathbf{m} = -\mathbf{m} \times \frac{\delta U[\mathbf{m}(x)]}{\delta \mathbf{m}(x)}. \tag{21}
$$

To obtain the functional derivative of the exchange energy (20), we compute its first variation:

$$
\delta \int dx \frac{A}{2}\left(\frac{d\mathbf{m}(x)}{dx}\right)^2 = \int dx A \frac{d\mathbf{m}(x)}{dx} \cdot \frac{d\delta\mathbf{m}(x)}{dx} = \int dx \left(-A\frac{d^2\mathbf{m}(x)}{dx^2}\right)\cdot \delta\mathbf{m}(x), \tag{22}
$$

whence $\delta U / \delta \mathbf{m}(x) = -A d^2 \mathbf{m}(x)/dx^2$. In the last step, we integrated by parts. The Landau–Lifshitz equation now reads

$$
\mathcal{S}\partial_t \mathbf{m} = A\mathbf{m} \times \partial_x^2 \mathbf{m}. \tag{23}
$$

### 3.2.1 Spin waves in a Heisenberg ferromagnet.

Spin waves near the uniform ground state $\mathbf{m}^{(0)} = (0, 0, 1)$ can be parametrized, as in the discrete case, by a complex field $\psi(x)$ so that $\mathbf{m} = (\operatorname{Re}\psi, \operatorname{Im}\psi, \sqrt{1 - |\psi|^2})$. Expanding Eq. (23) to the first order in $\psi$ yields the linear spin-wave equation

$$i\mathcal{S}\partial_t\psi = -A\partial_x^2\psi. \tag{24}$$

A harmonic wave, $\psi(t, x) = \psi(0, 0)e^{-i\omega t + ikx}$, has the spectrum

$$\omega = \frac{A}{\mathcal{S}}k^2 = |J|\mathcal{S}a^2k^2, \tag{25}$$

which agrees with our discrete result (14) in the long-wavelength limit $ka \ll 1$.

## 3.3 Scaling and symmetry considerations

The energy functional for the Heisenberg ferromagnetic chain (20) was derived directly from the lattice model (9). However, its general form can be deduced from very basic principles of scaling and symmetry.

*Scaling.* Speaking formally, the approximation of lattice variables $\mathbf{m}_n$ and their linear combinations by the field $\mathbf{m}(x)$ and its derivatives in Eq. (19) was organized as a Taylor expansion in powers of the operator $a\partial_x$, hence the name the *gradient expansion*. (This abstract notion becomes more tangible if we replace the gradient operator $\partial_x$ with the wavenumber $k$; then $ka \ll 1$ is obviously a small parameter in the long-wavelength limit.) When constructing a field theory, the rule of thumb is to keep only the lowest-order terms in the gradient expansion. Sometimes one may have to keep not just the leading-order term but also the next one.

*Symmetry.* Symmetries provide further restrictions on the possible forms of field theory. An energy functional (or an action) must remain invariant under symmetries of the physical system.

For example, the Heisenberg exchange interaction respects the symmetry of global spin rotations, $\mathbf{m}(x) \mapsto R\mathbf{m}(x)$, where $R$ is any $SO(3)$ matrix. Therefore, the exchange energy should depend on $\mathbf{m}(x)$ through a scalar quantity such as $\mathbf{m}\cdot\mathbf{m} \equiv \mathbf{m}^2$, which also happens to be of the lowest order in the gradient expansion. However, this quantity is trivial as $\mathbf{m}^2 = 1$, so we need to go to a higher order in the gradients. Terms of the first order in $\partial_x$ are ruled out if our magnet is symmetric under the inversion, $x \mapsto -x$. The second-order term $(\partial_x\mathbf{m})^2$ respects both the global spin rotations and inversion as well as the time-reversal symmetry $\mathbf{m}(x) \mapsto -\mathbf{m}(x)$. Hence the generic form of the exchange energy in a one-dimensional ferromagnet,

$$U[\mathbf{m}(x)] = \int dx \frac{A}{2}(\partial_x\mathbf{m})^2. \tag{26}$$

## 3.4 Easy-axis ferromagnet

Heisenberg exchange is the dominant form of spin interactions but not the only one. Relativistic spin-orbit coupling is a weaker interaction that breaks the symmetry of global $SO(3)$ spin rotations and brings the asymmetry of the atomic lattice to spins. In an atomic chain, the spatial direction along the chain is different from the other two (we assume the chain is embedded in our 3-dimensional space). Taking the $z$-axis along the chain, we find it plausible that the spins may favor the $z$ direction over $x$ and $y$. With the spin-rotation symmetry broken from $SO(3)$ to $SO(2)$ (rotations about the $z$ axis), we allow for terms like $m_z^2$ or $m_x^2 + m_y^2$ in the energy functional. This yields the simplest model of a ferromagnet with an easy axis:

$$U[\mathbf{m}(z)] = \int dz \left( \frac{A}{2}\mathbf{m}'^2 + \frac{K}{2}(m_x^2 + m_y^2) \right). \tag{27}$$

Here $K > 0$ is the anisotropy constant; the prime indicates the spatial derivative, $\mathbf{m}' \equiv d\mathbf{m}/dz$.

Note that the coupling constants $A$ and $K$ have different dimensions, J m and J/m, respectively. We may combine them to form a length scale $\lambda$ and an energy scale $\epsilon$:

$$\lambda = \sqrt{A/K}, \qquad \epsilon = \sqrt{AK}. \tag{28}$$

The weakness of the anisotropy relative to Heisenberg exchange means that the new length scale is large compared to the atomic lattice spacing, $\lambda \gg a$.

The easy-axis ferromagnet has two uniform ground states,

$$\mathbf{m}(z) = (0, 0, +1), \qquad \mathbf{m}(z) = (0, 0, -1). \tag{29}$$

They minimize the energy (27) absolutely, $U[\mathbf{m}(z)] = 0$.

In an infinite system, all finite-energy field configurations $\mathbf{m}(z)$ must approach one of the ground states as $z \to \pm\infty$. Thus they can be separated into 4 topological sectors distinguished by the pairs of values $\{m_z(-\infty), m_z(+\infty)\}$:

$$\{-1, -1\}, \quad \{+1, +1\}, \quad \{-1, +1\}, \quad \{+1, -1\}. \tag{30}$$

Configurations $\mathbf{m}(z)$ within the same topological sector are continuously deformable into one another. Configurations from different topological sectors are not because they have different boundary conditions. The minima of energy in the first two sectors are the uniform ground states. Energy minima in the last two are *domain walls*. They interpolate between the two ground states and have a positive energy. Domain walls owe their stability to their distinct topology: a domain-wall configuration cannot be continuously deformed into a uniform ground state.

### 3.4.1 Domain wall in an easy-axis ferromagnet

To find domain-wall solutions, we have to minimize the energy (27) subject to boundary conditions

$$\mathbf{m}(\pm\infty) = (0, 0, \pm\sigma), \quad \sigma = +1 \text{ or } -1. \tag{31}$$

This would seem to require the vanishing of the functional derivative, $\delta U/\delta \mathbf{m} = 0$. However, there is a technical complication here: not every variation $\delta \mathbf{m}$ is allowed, but only those that preserve the constraint on length, $|\mathbf{m}| = 1$. Rather than dealing with this constraint head-on, it pays to resolve it by expressing the three components of the unit vector $\mathbf{m}$ in terms of the polar angle $\theta$ and azimuthal angle $\phi$, see Eq. (A.4) in Appendix A. The energy is then expressed as a functional of two independent fields $\theta(z)$ and $\phi(z)$:

$$U[\theta(z), \phi(z)] = \int dz \left( \frac{A}{2} \left( \theta'^2 + \sin^2\theta \, \phi'^2 \right) + \frac{K}{2} \sin^2\theta \right). \tag{32}$$

Minimizing the energy with respect to fields $\phi$ and $\theta$ yields conditions

$$\begin{aligned} 0 &= \frac{\delta U}{\delta \phi} = -A(\sin^2\theta \, \phi')', \\ 0 &= \frac{\delta U}{\delta \theta} = -A\theta'' + (A\phi'^2 + K)\sin\theta \cos\theta. \end{aligned} \tag{33}$$

A uniform azimuthal angle, $\phi(z) = \Phi = \text{const}$, solves the first of Eqs. (33). The second one then reduces to a nonlinear differential equation $-A\theta'' + K\sin\theta \cos\theta = 0$ with a first integral

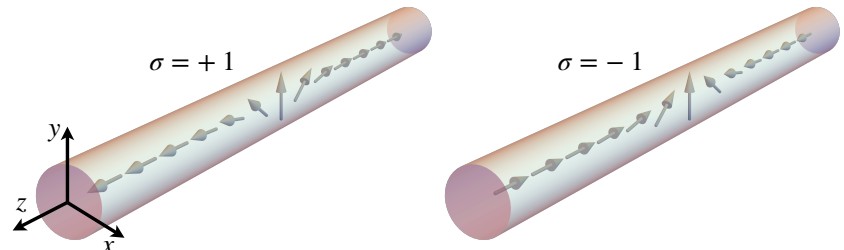

Figure 3: Domain walls in a uniaxial ferromagnet. Collective coordinates $Z$ and $\Phi$ quantify the translational and rotational degrees of freedom, respectively; $\sigma = \pm 1$ is a topological charge. After Ref. [1].

$A\theta'^2 - K\sin^2\theta = C = \text{const}$. The differential equation simplifies for $C = 0$: $\lambda\theta' = -\sigma\sin\theta$. We thus obtain domain-wall solutions, Fig. 3:

$$\cos\theta(z) = \sigma\tanh\frac{z-Z}{\lambda}, \quad \phi(z) = \Phi. \tag{34}$$

Here $\sigma = [m_z(+\infty) - m_z(-\infty)]/2 = \pm 1$ can be viewed as a *topological charge* of the domain wall. It plays an important role in the dynamics of the domain wall [1].

Integration constants $Z$ and $\Phi$ also have important roles. $Z$ specifies the location of the domain wall along the $z$ axis. $\Phi$ indicates the plane, in which the spin field interpolates between the two $z$-polarized ground states: for $\Phi = 0$, the spins lie in the $xz$ plane; for $\Phi = \pi/2$, they lie in the $yz$ plane; etc. These quantities are *collective coordinates* quantifying *zero modes*, i.e., motions that do not alter the energy of the domain wall, $U = 2\epsilon$. The existence of zero modes can be traced to the translational and rotational symmetries of the uniaxial ferromagnet.

When a domain wall is perturbed by a weak external force, it exhibits rigidity and retains its shape. The primary response is a change of the collective coordinates $Z$ and $\Phi$. The dynamics of collective coordinates is discussed in Ref. [1].

The simplified description of a uniaxial ferromagnet, offered by the field-theoretic approach, enabled us to readily find exact analytical solutions for a topological soliton, the domain wall. That would require substantially more effort in the original lattice model.

### 3.4.2 Dzyaloshinskii-Moriya term and helical magnetic order

If the atomic lattice lacks the inversion symmetry then the energy functional is allowed to have terms linear in the spatial gradients. Such terms also originate in the relativistic spin-orbit coupling and are therefore weak relative to the dominant exchange interactions. In an axially symmetric ferromagnetic chain, an axially symmetric term quadratic in the spin field would be proportional to the Lifshitz invariant $m_x\partial_z m_y - m_y\partial_z m_x$. Hence an updated version of the energy functional (27)

$$U[\mathbf{m}(z)] = \int dz \left(\frac{A}{2}\mathbf{m}'^2 - D(m_x m_y' - m_y m_x') + \frac{K}{2}(m_x^2 + m_y^2)\right). \tag{35}$$

The inversion-breaking Dzyaloshinskii–Moriya term (coupling constant $D$) can change the nature of the ground state of an easy-axis ferromagnet. If $D$ is strong enough, the ground state can be a magnetic helix. To see this, use the conical Ansatz,

$$\mathbf{m}(z) = (\sin\theta\cos kz, \sin\theta\sin kz, \cos\theta), \tag{36}$$

with a uniform polar angle $\theta$. Minimize the energy (35) with respect to the wavenumber $k$ and then the angle $\theta$. You will see that the ground state switches from $\theta = 0$ and $\pi$ (uniform $\mathbf{m}$) at weak $D$ to $\theta = \pi/2$ (helical $\mathbf{m}$) at strong $D$.

The general form of the gradient-linear term in ferromagnets with a broken inversion symmetry is

$$U_{\text{DM}}[\mathbf{m}] = \int dV\, \mathbf{D}_i \cdot (\mathbf{m} \times \partial_i \mathbf{m}), \tag{37}$$

with a separate Dzyaloshinskii vector $\mathbf{D}_i$ for each gradient direction $\partial_i$.

### 3.5 Historical note

The field theory of a ferromagnet and its application to domain walls was developed by Landau and Evgeny Lifshitz [2]. See Bar'yakhtar and Ivanov [3] for a historical account. The Dzyaloshinskii-Moriya term in the continuum version (37) was introduced phenomenologically by Dzyaloshinskii [4]. Moriya [5] derived a lattice version.

## 4 Field theory of a 2-sublattice antiferromagnet

### 4.1 Sublattice magnetizations

We return to the antiferromagnetic Heisenberg chain. What would be the right field theory? The approach used for the ferromagnetic case won't work: the assumption of slow spatial variation breaks down because in the ground state of the antiferromagnet adjacent spins point in opposite directions, $\mathbf{m}_{n+1} = -\mathbf{m}_n$.

To address this problem, we may introduce two separate spin fields $\mathbf{m}_1(x)$ and $\mathbf{m}_2(x)$ for the odd and even magnetic sublattices:

$$\mathbf{m}_n = \begin{cases} \mathbf{m}_1(na), & \text{if } n \text{ is odd,} \\ \mathbf{m}_2(na), & \text{if } n \text{ is even.} \end{cases} \tag{38}$$

What are the equations of motion for these fields? We have seen that the Landau–Lifshitz equation can be applied to a single spin (4), to individual spins in a lattice model (10), and to a spin field (21). The extension to two sublattice fields $\mathbf{m}_1$ and $\mathbf{m}_2$ is obvious:

$$\mathcal{S}\partial_t \mathbf{m}_1 = -\mathbf{m}_1 \times \frac{\delta U}{\delta \mathbf{m}_1}, \quad \mathcal{S}\partial_t \mathbf{m}_2 = -\mathbf{m}_2 \times \frac{\delta U}{\delta \mathbf{m}_2}. \tag{39}$$

Here $\mathcal{S}$ is the density of spins on one sublattice.

In low-energy states, adjacent spins are nearly antiparallel, and so are the sublattice fields, $\mathbf{m}_1(x) \approx -\mathbf{m}_2(x)$. It may be tempting to declare this description redundant as the two fields are nearly identical copies of each other—apart from the sign—and to simply replace $\mathbf{m}_2(x)$ with $-\mathbf{m}_1(x)$. However, this will only work for the ground states. It turns out that even in weakly excited states the two fields are not exactly antiparallel. We will follow a more systematic approach.

But first, a toy model that will give us some important insights.

### 4.2 Toy model: Two spins

Consider two spins $\mathbf{S}_1 = S\mathbf{m}_1$ and $\mathbf{S}_2 = S\mathbf{m}_2$ of the same length $S$ coupled by antiferromagnetic exchange, with potential energy $U = J\mathbf{S}_1 \cdot \mathbf{S}_2 = JS^2\mathbf{m}_1 \cdot \mathbf{m}_2$. Their equations of motion (10) read

$$S\dot{\mathbf{m}}_1 = -JS^2\mathbf{m}_1 \times \mathbf{m}_2, \qquad S\dot{\mathbf{m}}_2 = -JS^2\mathbf{m}_2 \times \mathbf{m}_1. \tag{40}$$

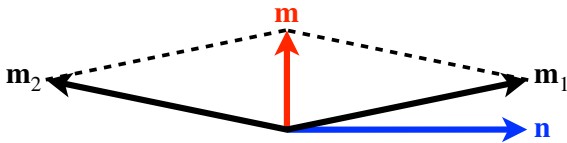

Figure 4: Net magnetization $\mathbf{m}$ and the Néel vector $\mathbf{n}$.

We introduce two new variables,

$$\mathbf{m} = \mathbf{m}_1 + \mathbf{m}_2\,, \qquad \mathbf{n} = \frac{\mathbf{m}_1 - \mathbf{m}_2}{2}\,, \tag{41}$$

as shown in Fig. 4. $S\mathbf{m}$ gives the total spin, or **m**agnetization. $\mathbf{n}$ is referred to as the **N**éel vector. At low energies, when $\mathbf{S}_1$ and $\mathbf{S}_2$ are close to antiparallel, $\mathbf{S}_1 \approx -\mathbf{S}_2 \approx S\mathbf{n}$, so the Néel vector approximates (up to a sign, maybe) both spins. Although net magnetization $\mathbf{m}$ is very small at low energies, we should not discard it as it plays an important role in the dynamics.

The equations of motion for the new vectors are

$$\dot{\mathbf{m}} = 0\,, \quad \dot{\mathbf{n}} = JS\mathbf{m} \times \mathbf{n}\,. \tag{42}$$

The net magnetization $\mathbf{m}$ is conserved, while the Néel vector $\mathbf{n}$ precesses around $\mathbf{m}$ at the frequency $\boldsymbol{\Omega} = JS\mathbf{m}$. We now see why it would be wrong to neglect $\mathbf{m}$: it facilitates the dynamics of the Néel vector. Adding weak interactions breaking the rotational symmetry will induce slow dynamics of the net magnetization $\mathbf{m}$; the Néel vector will then precess around the slowly moving $\mathbf{m}(t)$.

Note that the length constraints $\mathbf{m}_1^2 = \mathbf{m}_2^2 = 1$ translate into

$$\mathbf{m} \cdot \mathbf{n} = 0\,, \quad \mathbf{n}^2 + \frac{1}{4}\mathbf{m}^2 = 1\,. \tag{43}$$

### 4.3 Net and staggered magnetizations

We apply the lessons from the toy model to the antiferromagnetic chain and introduce the fields of net magnetization and Néel vector, or staggered magnetization:

$$\mathbf{m}(x) = \mathbf{m}_1(x) + \mathbf{m}_2(x)\,, \quad \mathbf{n}(x) = \frac{\mathbf{m}_1(x) - \mathbf{m}_2(x)}{2}\,. \tag{44}$$

The next step is to construct the appropriate energy functional. A term proportional to $\mathbf{m}^2$, allowed by symmetry and of the lowest order in the gradient expansion, expresses the tendency to suppress the net magnetization in an antiferromagnet. We omit higher-order gradient terms for $\mathbf{m}$. In contrast, staggered magnetization $\mathbf{n}$ has the length close to 1 and shouldn't be penalized by a term like $\mathbf{n}^2$. If the inversion symmetry is present, the gradient expansion should start at the second order. Hence the energy functional for an antiferromagnet:

$$U[\mathbf{m}(x), \mathbf{n}(x)] = \int dx \left( \frac{\mathbf{m}^2}{2\chi} + \frac{A}{2}(\partial_x \mathbf{n})^2 + \dots \right)\,. \tag{45}$$

Here $\chi$ can be viewed as the paramagnetic susceptibility. (To see that, add the Zeeman term $-\mathbf{m} \cdot \mathbf{h}$ to the energy and minimize it with respect to $\mathbf{m}$ to obtain $\mathbf{m} = \chi \mathbf{h}$.) The omitted terms are additional, weaker $\mathbf{n}$-dependent terms coming from spin-orbit coupling etc.

To derive the equations of motion for the net and staggered magnetization, we start with Eq. (39) and transform the functional derivatives as follows:

$$\frac{\delta}{\delta \mathbf{m}_{1,2}} = \frac{\delta}{\delta \mathbf{m}} \pm \frac{1}{2}\frac{\delta}{\delta \mathbf{n}}\,. \tag{46}$$

This yields the equations for $\mathbf{m}$ and $\mathbf{n}$:

$$
\begin{aligned}
\mathcal{S}\partial_t \mathbf{n} &= \frac{\mathbf{m}}{\chi} \times \left( \mathbf{n} - \frac{\chi}{4}\frac{\delta U}{\delta \mathbf{n}} \right) \approx \frac{\mathbf{m}}{\chi} \times \mathbf{n}, \\
\mathcal{S}\partial_t \mathbf{m} &= -\mathbf{n} \times \frac{\delta U}{\delta \mathbf{n}}.
\end{aligned}
\tag{47}
$$

As in the toy model of Sec. 4.2, the staggered magnetization precesses around the net magnetization at the angular frequency $\boldsymbol{\Omega} = \mathbf{m}/\chi\mathcal{S}$. (A small correction coming from the potential energy of $\mathbf{n}$ is small in weakly excited states and can therefore be neglected.) This result and the orthogonality constraint (43) allow us to express $\mathbf{m}$ in terms of $\mathbf{n}$:

$$
\mathbf{m} \approx \chi \mathcal{S} \mathbf{n} \times \partial_t \mathbf{n}.
\tag{48}
$$

The second of Eqs. (47) equates the local rate of change of the angular momentum to a conservative torque, which makes it the counterpart of the Landau–Lifshitz equation in a ferromagnet. By eliminating $\mathbf{m}$ in favor of $\mathbf{n}$, we obtain the analog of the Landau–Lifshitz equation for a 2-sublattice antiferromagnet:

$$
\partial_t(\rho \mathbf{n} \times \partial_t \mathbf{n}) = -\mathbf{n} \times \frac{\delta U}{\delta \mathbf{n}}.
\tag{49}
$$

Here $\rho \mathbf{n} \times \partial_t \mathbf{n} = \mathcal{S}\mathbf{m}$ is the local density of angular momentum and $\rho = \chi\mathcal{S}^2$ can be regarded as the moment of inertia for staggered magnetization. The left-hand side of Eq. (49) can be also written as $\rho \mathbf{n} \times \partial_t^2 \mathbf{n}$:

$$
\rho \mathbf{n} \times \partial_t^2 \mathbf{n} = -\mathbf{n} \times \frac{\delta U}{\delta \mathbf{n}}.
\tag{50}
$$

### 4.3.1 Spin waves in an antiferromagnetic Heisenberg chain

For the Heisenberg antiferromagnet (exchange only, no anisotropic intractions), the Landau–Lifshitz equation (50) reads

$$
\rho \mathbf{n} \times \partial_t^2 \mathbf{n} = A \mathbf{n} \times \partial_x^2 \mathbf{n}.
\tag{51}
$$

Small-amplitude spin waves near a ground state have a linear spectrum, $\omega = ck$, where the "speed of light" is $c = \sqrt{A/\rho}$. The linear spectrum is in agreement with our prior result for the spectrum of spin waves in an antiferromagnetic Heisenberg chain, Eq. (18).

## 4.4 Lagrangian

A path to quantization lies through a Lagrangian, whether we use canonical quantization or path integrals. We shall reverse-engineer the Lagrangian for the field of staggered magnetization $\mathbf{n}$ from its equation of motion (49).

One part of the Lagrangian is obvious, the potential energy $U[\mathbf{n}]$. The equation of motion (50) contains a second-order time derivative, which hints at the presence of kinetic energy. Indeed, the paramagnetic energy term $\mathbf{m}^2/2\chi$ in the energy functional (45) turns into a kinetic energy once we eliminate the hard field $\mathbf{m}$ in favor of $\mathbf{n}$ with the aid of Eq. (48):

$$
\frac{\mathbf{m}^2}{2\chi} = \frac{\chi\mathcal{S}^2(\mathbf{n} \times \partial_t \mathbf{n})^2}{2} \approx \frac{\chi\mathcal{S}^2(\partial_t \mathbf{n})^2}{2}.
\tag{52}
$$

In the last step, we approximated $\mathbf{n}$ as a unit vector, which leads to the orthogonality of $\mathbf{n}$ and $\partial_t \mathbf{n}$. We have thus guessed the general form of the Lagrangian for the field $\mathbf{n}$,

$$
L[\mathbf{n}] = \int dx\, \frac{\rho(\partial_t \mathbf{n})^2}{2} - U[\mathbf{n}] = \int dx \left( \frac{\rho(\partial_t \mathbf{n})^2}{2} - \frac{A(\partial_x \mathbf{n})^2}{2} - \dots \right).
\tag{53}
$$

Here $\rho = \chi \mathcal{S}^2$; the omitted terms may include weak anisotropic interactions induced by the relativistic spin-orbit coupling.

Let us derive the Landau–Lifshitz equation (50) from this Lagrangian. This is done by minimizing the action $S = \int L \, dt$ with respect to the field $\mathbf{n}$. For a minimal action, the first variation of the action $\delta S = 0$. The equation of motion is usually obtained from the condition $\delta S / \delta \mathbf{n} = 0$. However, there is a subtlety here: the field $\mathbf{n}$ is constrained to have the unit length, so we must be careful to avoid variations of $\mathbf{n}$ that change its length. This can be conveniently done by using Lagrange's method of undetermined multipliers.

To enforce the constraint $\mathbf{n}^2 = 1$ at every point in spacetime $(t, x)$, we modify the action as follows:

$$\tilde{S} = S - \int dt \, dx \, \frac{1}{2} \Lambda(t, x) \mathbf{n}^2(t, x). \tag{54}$$

Here $\Lambda(t, x)$ is a Lagrange multiplier. Minimization of the modified action yields the equation of motion

$$0 = \frac{\delta \tilde{S}}{\delta \mathbf{n}} = -\rho \, \partial_t^2 \mathbf{n} - \frac{\delta U}{\delta \mathbf{n}} - \Lambda \mathbf{n}. \tag{55}$$

The role of the Lagrange multiplier $\Lambda$ is to balance the longitudinal (i.e., parallel to $\mathbf{n}$) component of this equation. We do not need that component for transverse (tangential to the unit sphere) motion of $\mathbf{n}$. We can get rid of the longitudinal part by taking the cross product of Eq. (55) with $\mathbf{n}$. The term with the Lagrange multiplier then goes away as $\Lambda \mathbf{n} \times \mathbf{n} = 0$, and we obtain the Landau–Lifshitz equation (50).

### 4.4.1 "Lorentz" invariance

An important feature of the Lagrangian (53) is its "relativistic" form. The action is invariant under "Lorentz" transformations,

$$x \mapsto \frac{x - vt}{\sqrt{1 - v^2/c^2}}, \quad t \mapsto \frac{t - vx/c^2}{\sqrt{1 - v^2/c^2}}. \tag{56}$$

Note that the Lorentz invariance holds as long as the anisotropic potential $\mathcal{V}(\mathbf{n})$ depends on the vector $\mathbf{n}$ but not on its derivatives. Inclusion of the Dzyaloshinskii-Moriya term $\mathbf{D} \cdot (\mathbf{n} \times \partial_x \mathbf{n})$ would break this symmetry.

### 4.4.2 Domain wall in an easy-axis antiferromagnet

Let us look at an easy-axis antiferromagnetic chain. Lattice anisotropy, through the relativistic spin-orbit coupling, introduces anisotropy for spins. The appropriate potential term would be the same as for the ferromagnetic chain, Sec. 3.4.1, $\mathcal{V}(\mathbf{n}) = K(n_x^2 + n_y^2)/2$. Hence the potential energy functional

$$U[\mathbf{n}(z)] = \int dz \left( \frac{A}{2} \mathbf{n}'^2 + \frac{K}{2} (n_x^2 + n_y^2) \right). \tag{57}$$

Because the energy functional is exactly the same as it was for the ferromagnet (57), static domain-wall configurations $\mathbf{n}(z)$ can be read off from Eq. (34):

$$\mathbf{n}(z) = \left( \mathrm{sech} \, \frac{z - Z}{\lambda} \cos \Phi, \, \mathrm{sech} \, \frac{z - Z}{\lambda} \sin \Phi, \, \sigma \tanh \frac{z - Z}{\lambda} \right). \tag{58}$$

Here again $\lambda = \sqrt{A/K}$ is the spatial extent of the domain wall and $\sigma = \pm 1$ is a topological charge of the domain wall.

The "Lorentz" invariance of the Lagrangian (53) immediately allows us to construct solutions for a moving domain wall by simply replacing

$$z \mapsto \frac{z - vt}{\sqrt{1 - v^2/c^2}}, \tag{59}$$

in Eq. (58). Doing so yields a domain wall moving at velocity $v$. Note the "Lorentz" contraction of the domain wall, whose characteristic width shrinks from $\lambda$ to $\lambda\sqrt{1 - v^2/c^2}$! "Relativistic" effects in the dynamics of an antiferromagnetic domain wall were observed experimentally [6].

### 4.5 Net spin of an antiferromagnetic chain

#### 4.5.1 Classical ground state

Let us calculate the total spin of an antiferromagnetic chain. That this task is surprisingly nontrivial can be appreciated from a very simple example, the classical ground state, in which

$$\mathbf{S}_n = (-1)^{n+1} S \mathbf{n}, \tag{60}$$

where $\mathbf{n}$ is a constant vector of staggered magnetization. In a chain with an even number of spins $2N$,

$$\mathbf{S} = S\mathbf{n} \sum_{n=1}^{2N} (-1)^{n+1} = 0. \tag{61}$$

However, if the chain has an odd number of spins $2N + 1$ then

$$\mathbf{S} = S\mathbf{n} \sum_{n=1}^{2N+1} (-1)^{n+1} = S\mathbf{n}. \tag{62}$$

This is not particularly surprising, of course. In a chain with $2N$ spins, half of them are $S\mathbf{n}$ and the other half $-S\mathbf{n}$, so the total is zero. Adding one more spin $S\mathbf{n}$ to the chain raises the total to $S\mathbf{n}$. For a chain that begins and ends with a spin on sublattice 2,

$$\mathbf{S} = S\mathbf{n} \sum_{n=0}^{2N} (-1)^{n+1} = -S\mathbf{n}. \tag{63}$$

We see that the seemingly simple question *What is the net spin of an antiferromagnetic chain?* does not have a unique answer even for the classical ground state. It could be 0 if the chain has an even number of sites or $S\mathbf{n}$ if the number of sites is odd. Furthermore, in the latter case it could also be $-S\mathbf{n}$ if both end spins are on sublattice 2.

We can fix the problem by agreeing to work with chains containing an even number of spins. Then $\mathbf{S} = 0$ in a ground state.

#### 4.5.2 Domain wall: Wrong answer

Consider next a more complex example, a spin chain with a domain wall. We used a continuum theory to obtain an analytical solution in Sec. 4.4.2, so let us compute the total spin in the same language:

$$\mathbf{S} = S\left[\mathbf{m}_1(a) + \mathbf{m}_2(2a) + \mathbf{m}_1(3a) + \mathbf{m}_2(4a) + \ldots + \mathbf{m}_1(2Na - a) + \mathbf{m}_2(2Na)\right]. \tag{64}$$

It is tempting to lump together adjacent spins to obtain $\mathbf{m}_1(a) + \mathbf{m}_2(2a) \approx \mathbf{m}(x)$, where $x = 3a/2$ and write the total spin in terms of the uniform magnetization field $\mathbf{m}$:

$$\mathbf{S} \approx S \sum_{n=1}^{N} \mathbf{m}(2na - a/2) = \frac{S}{2a} \sum_{n=1}^{N} \mathbf{m}(2na - a/2)\, 2a \approx \int dx\, S\mathbf{m}(x). \tag{65}$$

The uniform magnetization field $\mathbf{m}$ is then expressed in terms of the staggered field $\mathbf{n}$ through Eq. (48) to obtain

$$\mathbf{S} \approx \int dx\, \rho\, \mathbf{n} \times \partial_t \mathbf{n}. \tag{66}$$

This result gives the correct answer—zero—for a classical ground state, in which the spins are static, $\partial_t \mathbf{n} = 0$. The same logic applies to a static domain wall, or any static soliton, so they also have $\mathbf{S} = 0$.

It turns out that our hasty calculation is not quite right. We shall perform a more careful analysis next.

### 4.5.3 Domain wall: Right answer

Let us do a more careful job. Suppose our chain begins with a site on sublattice 1 with cordinate $x_1 = a$ and ends with a site on sublattice 2 with coordinate $x_{2N} = 2Na$. Expressing the discrete spin variables in terms of the sublattice magnetization fields yields

$$\mathbf{S}_{2n-1} = S\mathbf{m}_1(x_{2n-1}) = \frac{S}{2}\mathbf{m}(x_{2n-1}) + S\mathbf{n}(x_{2n-1}), \tag{67}$$

for a site on sublattice 1. A site on sublattice 2 is shifted by $a$ to the right, $x_{2n} = x_{2n-1} + a$, so

$$\mathbf{S}_{2n} = S\mathbf{m}_2(x_{2n-1} + a) = \frac{S}{2}\mathbf{m}(x_{2n-1} + a) - S\mathbf{n}(x_{2n-1} + a). \tag{68}$$

We use the Taylor expansion and keep only the lowest-order term:

$$\mathbf{S}_{2n} = \frac{S}{2}[\mathbf{m}(x_{2n-1}) + \partial_x m(x_{2n-1})a + \ldots] - S[\mathbf{n}(x_{2n-1}) + \partial_x \mathbf{n}(x_{2n-1})a + \ldots]. \tag{69}$$

In the spirit of the preceding sections, we keep the gradient terms for $\mathbf{n}$ and drop them for $\mathbf{m}$ and pass from the summation to integration to obtain

$$\mathbf{S} = \sum_{n=1}^{N}(\mathbf{S}_{2n-1} + \mathbf{S}_{2n}) \approx \int dx\, S\mathbf{m}(x) - \frac{S}{2}\int dx\, \partial_x \mathbf{n}(x). \tag{70}$$

Comparing this expression to our previous result (65), we see that a nonuniform staggered magnetization field may contribute to the net spin. This contribution has an interesting character: it is *topological*. Integrating the spatial-derivative term shows that the correction depends on the difference of the staggered magnetization at the ends but not on its values in between. For an infinite chain,

$$\mathbf{S} = \int dx\, \rho\, \mathbf{n} \times \partial_t \mathbf{n} - \frac{S}{2}\mathbf{n}(x)\Big|_{-\infty}^{+\infty}. \tag{71}$$

A domain wall reverses the direction of staggered magnetization, $\mathbf{n}(-\infty) = -\mathbf{n}(+\infty)$. A static domain wall then has the net spin

$$\mathbf{S} = -S\mathbf{n}(+\infty). \tag{72}$$

This result is topological in the sense that it is sensitive not to the precise *geometry* of a domain wall (i.e., its shape) but only to its *topology* (i.e., which ground states it interpolates between).

Repeating the same analysis for a chain that begins with a site of sublattice 2 and ends with a site of sublattice 1 yields an answer with the opposite sign of the topological term:

$$\mathbf{S} = \int dx\, \rho\, \mathbf{n} \times \partial_t \mathbf{n} + \frac{S}{2}\mathbf{n}(x)\Big|_{-\infty}^{+\infty}. \tag{73}$$

Then a static domain wall has the net spin

$$\mathbf{S} = S\mathbf{n}(+\infty).\tag{74}$$

Although we obtained this result in the framework of classical physics, it also applies to quantum spins. Faddeev and Takhtajan [7] showed that elementary excitations of the spin-1/2 antiferromagnetic Heisenberg chain are particles carrying spin 1/2. These particles, known nowadays as *spinons*, are quantum solitons related to domain walls. For classical spins, the topological contribution to the net spin of a domain wall was obtained by Papanicolaou [8].

### 4.6 Historical note

The dynamics of an antiferromagnet in the Lagrangian form was first formulated by Bar'yakhtar and Ivanov [9]. Mikeska [10] and Haldane [11] used the Hamiltonian approach.

## 5 Field theory of a 3-sublattice antiferromagnet

We now turn to antiferromagnets with 3 magnetic sublattices, arising in magnets with non-bipartite lattice geometry such as triangular, Fig. 1(c), and kagome, Fig. 1(d). In contrast to ferromagnets and simple antiferromagnets, where the field theory could be written in terms of a single vector, the field theory of a 3-sublattice antiferromagnet requires a more complex language.

The elementary building block of these magnets is a triangle with three spins $\mathbf{S}_1 = S\mathbf{m}_1$, $\mathbf{S}_2 = S\mathbf{m}_2$, and $\mathbf{S}_3 = S\mathbf{m}_3$, where $S$ is the spin length and $\mathbf{m}_i$ are unit vectors. The exchange energy of this building block can be written as

$$U = \frac{JS^2}{2}(\mathbf{m}_1 + \mathbf{m}_2 + \mathbf{m}_3)^2.\tag{75}$$

In a ground state, $\mathbf{m}_1 + \mathbf{m}_2 + \mathbf{m}_3 = 0$, which means that the spins are coplanar and point at angles of 120° relative to one another. It is clear that a single vector is not enough to describe such states.

Three spins locked in a 120° coplanar arrangement can be regarded as a *rigid body*. The orientation of a rigid body can be specified in a number of ways. Perhaps the most familiar parametrization is in terms of three Euler rotations starting with a reference orientation. It is possible to formulate the field theory in terms of the three Euler angles, similarly to what we did with two spherical angles in Sec. 3.4.1. The drawback of this approach is that it hides the spin rotational symmetry of the Heisenberg model. Nonetheless, in some situations it can be useful.

Dombre and Read [12] parametrized their field theory of a 3-sublattice antiferromagnet in terms of $SO(3)$ matrices, which can also be used to specify the orientation of a rigid body. $SO(3)$ matrices are rather abstract objects, so it is hard to build intuition about them. Furthermore, the field theory of Dombre and Read was formulated for the triangular lattice, which turns out to be a special case. In this section, I will describe a field theory of a 3-sublattice antiferromagnet that applies more broadly [13].

### 5.1 Toy model: Three spins

To see the basic outlines of our approach, consider the dynamics of the basic building block, the three spins coupled by antiferromagnetic exchange (75). The equations of motion of the three spins are

$$S\dot{\mathbf{m}}_\alpha = -\mathbf{m}_\alpha \times \frac{\partial U}{\partial \mathbf{m}_\alpha} = -\mathbf{m}_\alpha \times JS^2\mathbf{m},\tag{76}$$

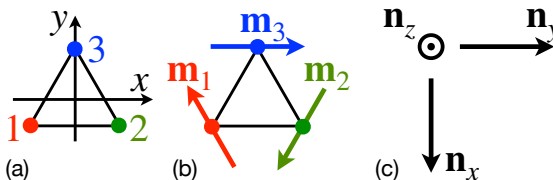

Figure 5: (a) Lattice axes $x$ and $y$. (b) A ground state of three spins $S\mathbf{m}_1$, $S\mathbf{m}_2$, and $S\mathbf{m}_3$ on a triangle. (b) The corresponding spin-frame vectors $\mathbf{n}_x$, $\mathbf{n}_y$, and $\mathbf{n}_z$.

where $\alpha = 1, 2, 3$, and

$$\mathbf{m} = \mathbf{m}_1 + \mathbf{m}_2 + \mathbf{m}_3 \,, \tag{77}$$

is (up to a factor $S$) the net spin. We can see that all three spins precess about the vector of net spin at the frequency $\boldsymbol{\Omega} = JS\mathbf{m}$.

Let us introduce two more vectors, which will serve as analogs of the staggered magnetization $\mathbf{n}$ for the simple antiferromagnet:

$$\mathbf{n}_x = \frac{\mathbf{m}_2 - \mathbf{m}_1}{\sqrt{3}} \,, \quad \mathbf{n}_y = \frac{2\mathbf{m}_3 - \mathbf{m}_2 - \mathbf{m}_1}{3} \,. \tag{78}$$

In a ground state, these new vectors have unit length and are orthogonal to each other, Fig. 5.

These new vectors are linear combinations of the three spin vectors, so they also precess about the net spin direction at the frequency $\boldsymbol{\Omega} = JS\mathbf{m}$:

$$\dot{\mathbf{n}}_a = JS\mathbf{m} \times \mathbf{n}_a \,, \tag{79}$$

where $a = x, y$.

This toy model gives us a familiar message. Although the net spin $\mathbf{m}$ is suppressed by antiferromagnetic exchange, it plays an important role, mediating the dynamics of the more visible variables, the staggered magnetizations $\mathbf{n}_x$ and $\mathbf{n}_y$.

What about the labels $x$ and $y$ for the two staggered magnetizations? They reflect symmetry properties of the new vectors under transformations of the point group of the triangle $D_3$. These transformations, known as *exchange symmetries* [14], do not affect the spin orientations and only exchange the vertices of the triangle. For example, under a $C_2$ rotation about the $y$ axis of the triangle, Fig. 5(a), leads to a permutation of the site labels, $\{1, 2, 3\} \mapsto \{2, 1, 3\}$, and therefore to a similar permutation of the spin variables: $\{\mathbf{m}_1, \mathbf{m}_2, \mathbf{m}_3\} \mapsto \{\mathbf{m}_2, \mathbf{m}_1, \mathbf{m}_3\}$. To repeat, the spins are moved to new spatial positions without rotating. The staggered magnetizations transform in the following way: $\{\mathbf{n}_x, \mathbf{n}_y\} \mapsto \{-\mathbf{n}_x, \mathbf{n}_y\}$, precisely as spatial coordinates would transform under that $C_2$ rotation, $\{x, y\} \mapsto \{-x, y\}$, hence the labels. It can be checked that $\{\mathbf{n}_x, \mathbf{n}_y\}$ transform in terms of each other like $\{x, y\}$ do under all symmetries of point group $D_3$.

The message from the toy model is that staggered magnetizations $\mathbf{n}_a$ ($a = x, y$) behave as 3-dimensional vectors under global spin rotations and as Cartesian components of a 2-dimensional vector under point-group operations, in the same way as, say, gradients $\partial_a$ do. This means, for example, that $\partial_a\mathbf{n}_a$ (with summation over $a$ is implied) is a spin vector and a spatial scalar.

## 5.2 Net and staggered magnetization fields

Our field theory starts with three magnetization fields $\mathbf{m}_1(x, y)$, $\mathbf{m}_2(x, y)$, and $\mathbf{m}_3(x, y)$ for each magnetic sublattice. We then introduce the fields of net magnetization $\mathbf{m}$ and two staggered magnetizations $\mathbf{n}_x$ and $\mathbf{n}_y$ in the same wasy as we did for three spins in Eqs. (77)

and (78):

$$\mathbf{m} = \mathbf{m}_1 + \mathbf{m}_2 + \mathbf{m}_3 \,, \quad \mathbf{n}_x = \frac{\mathbf{m}_2 - \mathbf{m}_1}{\sqrt{3}} \,, \quad \mathbf{n}_y = \frac{2\mathbf{m}_3 - \mathbf{m}_2 - \mathbf{m}_1}{3} \,. \tag{80}$$

Uniform magnetization $\mathbf{m}$ is strongly suppressed by the Heisenberg exchange, so $\mathbf{m}^2 \ll 1$ in low-energy states. The staggered fields $\mathbf{n}_x$ and $\mathbf{n}_y$ are mutually orthogonal and have unit length (an approximation that becomes exact in a ground state). With the symmetry properties of the staggered fields clarified in Sec. 5.1, we are now in position to build the field theory.

## 5.3 Potential energy

We begin with potential energy. For any lattice with hexagonal symmetry, the potential energy functional should be symmetric under time reversal, lattice translations, global spin rotations and operations of point group $D_3$. Suppression of the net magnetization by exchange is obviously represented by a paramagnetic term $\mathbf{m}^2/2\chi$, with $\chi$ being the paramagnetic susceptibility, just like for a 2-sublattice antiferromagnet.

Next, in analogy with a 2-sublattice antiferromagnet, we should add terms quadratic in both spatial gradients $\partial_a$ and staggered magnetizations $\mathbf{n}_a$. Such terms will be invariant under time reversal and lattice translations. Taking a scalar product between the participating $\mathbf{n}_a$ vectors would ensure invariance under global spin rotations.

To take care of the remaining symmetry—point-group operations—we temporarily enlarge it from $D_3$ to $D_\infty$, the dihedral group that includes all possible spatial rotations in the $xy$ plane and $C_2$ rotations about axes lying in the $xy$ plane. We have narrowed down the possible terms to linear combinations of $\partial_a \mathbf{n}_b \cdot \partial_c \mathbf{n}_d$. These quantities can be thought of as components of a fourth-rank tensor with respect to group $D_\infty$. To turn them into $D_\infty$ scalars, we simply make pairwise contractions of the indices, e.g., $\partial_a \mathbf{n}_a \cdot \partial_c \mathbf{n}_c$. Such terms are invariant under all operations of $D_\infty$, and therefore of its subgroup $D_3$.

We thus arrive at the potential energy for a Heisenberg antiferromagnet with 3 sublattices:

$$U[\mathbf{m}, \mathbf{n}_x, \mathbf{n}_y] = \int d^2 r \left( \frac{\mathbf{m}^2}{2\chi} + \frac{\lambda}{2} \partial_a \mathbf{n}_a \cdot \partial_b \mathbf{n}_b + \frac{\mu}{2} \partial_a \mathbf{n}_b \cdot \partial_a \mathbf{n}_b + \frac{\nu}{2} \partial_a \mathbf{n}_b \cdot \partial_b \mathbf{n}_a \right) . \tag{81}$$

Three ways of contracting the Cartesian indices of $\partial_a \mathbf{n}_b \cdot \partial_c \mathbf{n}_d$ pairwise result in three terms with coupling constants $\lambda$, $\mu$, and $\nu$. These are invariants of the $D_\infty$ group, and therefore of its subgroup $D_3$.

There is a danger that, by relying on the invariants of $D_\infty$, we missed some of the invariants unique to $D_3$. Fortunately, such invariants can be formed from third-rank tensors, but not fourth-rank ones. Furthermore, third-rank tensors would contain an odd number of gradients or staggered fields and be odd under inversion or time reversal. So, at least at this order in the gradients and staggered magnetizations, we have all the terms allowed by symmetries.

## 5.4 Lagrangian

Potential energy is not enough to determine the dynamics of the system. We need the Lagrangian, which may include, in addition to the potential energy, kinetic and Berry-phase terms.

To derive the Lagrangian for our field theory, we recall the lesson of the toy model of Sec. 5.1: all vectors precess around the direction of net magnetization, Eq. (79). This result can be reproduced in the field theory if we write down the dynamics of the magnetization fields and only use the first term, proportional to $\mathbf{m}^2$, in Eq. (81) in the Landau–Lifshitz equations:

$$\mathcal{S} \partial_t \mathbf{m}_\alpha = -\mathbf{m}_\alpha \times \frac{\delta U}{\delta \mathbf{m}_\alpha} \approx -\mathbf{m}_\alpha \times \frac{\mathbf{m}}{\chi} \,, \quad \alpha = 1, 2, 3 \,. \tag{82}$$

At this level of approximation, all three sublattice magnetizations precess around the net magnetization at the frequency

$$\boldsymbol{\Omega} = \mathbf{m}/\chi\mathcal{S}. \tag{83}$$

The same applies to the net and staggered magnetizations, which are but their linear combinations:

$$\dot{\mathbf{m}} \approx \boldsymbol{\Omega} \times \mathbf{m}, \quad \dot{\mathbf{n}}_a \approx \boldsymbol{\Omega} \times \mathbf{n}_a, \quad a = x, y. \tag{84}$$

This approximation is sufficient for staggered magnetizations, but it is too crude for $\mathbf{m}$ because it yields no dynamics for it: $\dot{\mathbf{m}} \approx \mathbf{m} \times \mathbf{m}/\chi\mathcal{S} = 0$. The motion of uniform magnetization is induced by the neglected gradient terms in the potential energy (81). Restoring them yields an equation of motion highly reminiscent of the Landau–Lifshitz equation and its analogs, Eqs. (4), (21), (47):

$$\mathcal{S}\dot{\mathbf{m}} = -\mathbf{n}_a \times \frac{\delta U}{\delta \mathbf{n}_a}. \tag{85}$$

A reminder: summation is implied over the doubly repeated index $a = x, y$.

We could combine equations (84) and (85) to eliminate $\mathbf{m}$ and deduce the equations of motion for $\mathbf{n}_a$. Instead, we will pursue the more robust Lagrangian approach. A plausible Lagrangian for the fields $\mathbf{m}$ and $\mathbf{n}_a$ is

$$L[\mathbf{m}, \mathbf{n}_x, \mathbf{n}_y] = \int d^2 r \left( \mathcal{S}\boldsymbol{\Omega} \cdot \mathbf{m} - \frac{\mathbf{m}^2}{2\chi} - \frac{\lambda}{2}\partial_a \mathbf{n}_a \cdot \partial_b \mathbf{n}_b - \frac{\mu}{2}\partial_a \mathbf{n}_b \cdot \partial_a \mathbf{n}_b - \frac{\nu}{2}\partial_a \mathbf{n}_b \cdot \partial_b \mathbf{n}_a \right). \tag{86}$$

Indeed, it has the potential part given by Eq. (81) and its equation of motion for $\mathbf{m}$ yields the expected precession frequency (83).

The first term in the Lagrangian, $\mathcal{S}\boldsymbol{\Omega} \cdot \mathbf{m}$, is linear in the precession frequency, which is proportional to the velocities of the staggered magnetizations. This term then likely represents the spin Berry phase. It should actually be expressed in terms of the velocities $\dot{\mathbf{n}}_a$, so for the moment our Lagrangian (86) remains half-baked. We will fix this problem below.

For now, we proceed to integrate out the subdominant field of net magnetization. This is easy to do with the aid of its equation of motion (83). Elimination of $\mathbf{m}$ in favor of $\boldsymbol{\Omega}$ in Eq. (86) yields a new Lagrangian for staggered magnetizations alone:

$$L[\mathbf{n}_x, \mathbf{n}_y] = \int d^2 r \left( \frac{\rho \boldsymbol{\Omega}^2}{2} - \frac{\lambda}{2}\partial_a \mathbf{n}_a \cdot \partial_b \mathbf{n}_b - \frac{\mu}{2}\partial_a \mathbf{n}_b \cdot \partial_a \mathbf{n}_b - \frac{\nu}{2}\partial_a \mathbf{n}_b \cdot \partial_b \mathbf{n}_a \right). \tag{87}$$

The first term $\rho \boldsymbol{\Omega}^2/2$ is the kinetic energy of staggered magnetizations with the moment of inertia $\rho = \chi\mathcal{S}^2$, the same as for a 2-sublattice antiferromagnet (Sec. 4.3).

To express the local precession frequency $\boldsymbol{\Omega}$ in terms of the velocities of staggered magnetizations, we introduce a new field, known as the *vector spin chirality* [15],

$$\mathbf{n}_z = \frac{2}{3\sqrt{3}}(\mathbf{m}_1 \times \mathbf{m}_2 + \mathbf{m}_2 \times \mathbf{m}_3 + \mathbf{m}_3 \times \mathbf{m}_1). \tag{88}$$

In a ground state, when $\mathbf{m} = 0$, the three vectors $\mathbf{n}_i$, where $i = x, y, z$, form an orthonormal spin frame:

$$\mathbf{n}_i \cdot \mathbf{n}_j = \delta_{ij}, \quad \mathbf{n}_i \times \mathbf{n}_j = \epsilon_{ijk}\mathbf{n}_k. \tag{89}$$

Under spatial transformations of point group $D_3$, chirality vector $\mathbf{n}_z$ transforms as the $z$ component of a spatial vector, hence the label. Completeness of the basis $\{\mathbf{n}_x, \mathbf{n}_y, \mathbf{n}_z\}$ means that any spin vector can be expressed in terms of these basis vectors. E.g.,

$$\boldsymbol{\Omega} = \mathbf{n}_i(\mathbf{n}_i \cdot \boldsymbol{\Omega}). \tag{90}$$

As usual, summation is implied over the repeated index $i = x, y, z$. The spin frame rigidly rotates at the angular frequency $\mathbf{\Omega}$, $\dot{\mathbf{n}}_i = \mathbf{\Omega} \times \mathbf{n}_i$. It follows then that

$$\mathbf{n}_i \times \dot{\mathbf{n}}_i = \mathbf{n}_i \times (\mathbf{\Omega} \times \mathbf{n}_i) = \mathbf{\Omega}(\mathbf{n}_i \cdot \mathbf{n}_i) - \mathbf{n}_i(\mathbf{n}_i \cdot \mathbf{\Omega}) = 2\mathbf{\Omega}, \tag{91}$$

where we used the conditions of orthonormality (89) and completeness (90) for the spin frame. We have thus expressed the rotation frequency in terms of the velocities of the spin-frame vectors $\mathbf{n}_i$. Note that staggered magnetizations $\mathbf{n}_x$ and $\mathbf{n}_y$ alone would not be enough for this, so we had to complete the spin frame with chirality $\mathbf{n}_z$. The square of the angular velocity can be expressed as the kinetic energy of the spin-frame vectors:

$$\mathbf{\Omega}^2 = \frac{1}{2}\dot{\mathbf{n}}_i \cdot \dot{\mathbf{n}}_i. \tag{92}$$

The proof is left as an exercise for the reader.

With this, we have our final expression for the Lagrangian of the 3-sublattice Heisenberg antiferromagnet expressed in terms of the three spin-frame fields $\mathbf{n}_i$, $i = x, y, z$:

$$L[\mathbf{n}_x, \mathbf{n}_y, \mathbf{n}_z] = \int d^2r \left( \frac{\rho}{4}\partial_t \mathbf{n}_i \cdot \partial_t \mathbf{n}_i - \frac{\lambda}{2}\partial_a \mathbf{n}_a \cdot \partial_b \mathbf{n}_b - \frac{\mu}{2}\partial_a \mathbf{n}_b \cdot \partial_a \mathbf{n}_b - \frac{\nu}{2}\partial_a \mathbf{n}_b \cdot \partial_b \mathbf{n}_a \right). \tag{93}$$

A reminder: indices at the beginning of the Latin alphabet, $a, b, c, \ldots$, take on values $x$ and $y$, whereas those from the middle, $i, j, k, \ldots$, run through $x$, $y$, and $z$.

## 5.5 Equation of motion for a 3-sublattice antiferromagnet

Obtaining equations of motion from a Lagrangian is a straightforward task. However, there is an obstruction in our path: the three vectors $\mathbf{n}_i$ are not really independent in view of the constraints on their orthonormality (89). Therefore, we cannot vary them independently of one another.

To resolve the constraints, we rely on the method of Lagrange multipliers along the lines of Sec. 4.4. For each constraint $\mathbf{n}_i \cdot \mathbf{n}_j = \delta_{ij}$, we introduce a Lagrange multiplier $\Lambda_{ij} = \Lambda_{ji}$ and use a modified Lagrangian,

$$\tilde{L}[\mathbf{n}_x, \mathbf{n}_y, \mathbf{n}_z] = L[\mathbf{n}_x, \mathbf{n}_y, \mathbf{n}_z] - \int d^2r \frac{\Lambda_{ij}}{2}\mathbf{n}_i \cdot \mathbf{n}_j$$
$$= \int d^2r \left( \frac{\rho}{4}\partial_t \mathbf{n}_i \cdot \partial_t \mathbf{n}_i - \frac{\Lambda_{ij}}{2}\mathbf{n}_i \cdot \mathbf{n}_j \right) - U[\mathbf{n}_x, \mathbf{n}_y, \mathbf{n}_z], \tag{94}$$

where the potential energy $U[\mathbf{n}_x, \mathbf{n}_y, \mathbf{n}_z]$ includes the gradient terms for the pure Heisenberg model and possibly additional anisotropic terms.

Now we can minimize the resulting action $\tilde{S} = \int dt\, \tilde{L}$ with respect to the fields $\mathbf{n}_i$ as if they were independent. The resulting equation of motion for field $\mathbf{n}_i$ reads

$$0 = \frac{\delta\tilde{S}}{\delta\mathbf{n}_i} = -\frac{1}{2}\rho\partial_t^2 \mathbf{n}_i - \frac{\delta U}{\delta\mathbf{n}_i} - \Lambda_{ij}\mathbf{n}_j. \tag{95}$$

The Lagrange multipliers can now be removed by taking a cross product with $\mathbf{n}_i$ and summing over the doubly repeated index $i$. Because $\Lambda_{ij}$ is symmetric in its indices and $\mathbf{n}_i \times \mathbf{n}_j$ is antisymmetric, $\Lambda_{ij}\mathbf{n}_i \times \mathbf{n}_j = 0$.

We thus obtain the analog of the Landau–Lifshitz equation for the spin frame:

$$\frac{\rho}{2}\mathbf{n}_i \times \partial_t^2 \mathbf{n}_i = -\mathbf{n}_i \times \frac{\delta U}{\delta\mathbf{n}_i}, \tag{96}$$

where the index $i = x, y, z$ is summed over. The left-hand side can be expressed in terms of the angular frequency of precession:

$$\rho \partial_t \boldsymbol{\Omega} = -\mathbf{n}_i \times \frac{\delta U}{\delta \mathbf{n}_i} \tag{97}$$

[Hint: use the orthogonality of $\mathbf{n}_i$ and $\partial_t \mathbf{n}_i$ and the relation between $\boldsymbol{\Omega}$ and $\dot{\mathbf{n}}_i$ (91)].

Eqs. (96) and (97) have the structure common to classical equations of motion for magnets. The left-land side is the rate of change of the angular momentum density $\frac{1}{2}\rho \mathbf{n}_i \times \dot{\mathbf{n}}_i = \rho \boldsymbol{\Omega} = \mathcal{S}\mathbf{m}$, whereas the right-hand side is the conservative torque. Note that all 3 spin-frame vectors $\mathbf{n}_x$, $\mathbf{n}_y$, and $\mathbf{n}_z$ can contribute to the conservative torque. Our earlier cavalier derivation led us to Eq. (85), which omitted $\mathbf{n}_z$. It would not matter for the Heisenberg model, which has no energy dependence on $\mathbf{n}_z$, but generally this omission could lead to errors. This is why the formal Lagrangian route was preferable.

These vector equations have three components, which matches the number of independent fields in the problem. In the spin-frame representation, we have 9 components of the 3 unit vectors and 6 constraints $\mathbf{n}_i \cdot \mathbf{n}_j = \delta_{ij}$. Alternatively, we have 3 fields of the Euler angles. Thus, Eqs. (96) and (97) provide a maximally economical formulation of the equations of motion and at the same time retain the spin-rotational symmetry of the Heisenberg model.

For the Heisenberg model, Eq. (97) yields

$$\rho \partial_t \boldsymbol{\Omega} = \mathbf{n}_a \times \left[ (\lambda + \nu)\partial_a \partial_b \mathbf{n}_b + \mu \partial_b \partial_b \mathbf{n}_a \right]. \tag{98}$$

Observe that the exchange coupling constants $\lambda$ and $\nu$ enter the equation of motion not individually but as the sum $\lambda + \nu$. This happens because the potential terms $\partial_a \mathbf{n}_a \cdot \partial_b \mathbf{n}_b$ and $\partial_a \mathbf{n}_b \cdot \partial_b \mathbf{n}_a$ in the Lagrangian (93) are related by partial integration, so they produce the same effect in the bulk.

### 5.5.1 Spin waves in a 3-sublattice Heisenberg antiferromagnet

Spin waves near a uniform ground state $\mathbf{n}_i$ can be represented as small rotations of the spin frame, $\delta \mathbf{n}_i = \delta \boldsymbol{\phi} \times \mathbf{n}_i$, where $\delta \boldsymbol{\phi}(t, x, y)$ is an infinitesimal local angle of rotation. Expanding Eq. (98) to the first order in $\delta \boldsymbol{\phi}$ yields the linear spin-wave equation

$$\rho \partial_t^2 \delta \boldsymbol{\phi} = (\lambda + \nu)\mathbf{n}_a \times (\partial_a \partial_b \delta \boldsymbol{\phi} \times \mathbf{n}_b) + \mu \mathbf{n}_a \times (\partial_b \partial_b \delta \boldsymbol{\phi} \times \mathbf{n}_a). \tag{99}$$

We assume a harmonic spin wave with a frequency $\omega$ and wavevector $\mathbf{k} = (k, 0)$. Then the differential wave equation reduces to an algebraic one,

$$\rho \omega^2 \delta \boldsymbol{\phi} = (\lambda + \nu)k^2 \mathbf{n}_x \times (\delta \boldsymbol{\phi} \times \mathbf{n}_x) + \mu k^2 \mathbf{n}_a \times (\delta \boldsymbol{\phi} \times \mathbf{n}_a). \tag{100}$$

It has three eigenmodes, with polarizations $\delta \boldsymbol{\phi}_\mathrm{I} = \delta \phi \, \mathbf{n}_x$, $\delta \boldsymbol{\phi}_\mathrm{II} = \delta \phi \, \mathbf{n}_y$, and $\delta \boldsymbol{\phi}_\mathrm{III} = \delta \phi \, \mathbf{n}_z$. For the first of these, Eq. (100) reduces to $\rho \omega^2 \delta \boldsymbol{\phi} = \mu k^2 \delta \boldsymbol{\phi}$. This spin wave has the dispersion $\omega = \sqrt{\mu/\rho}\, k$. The other two polarizations are analyzed in a similar way. The three spin-wave modes have propagation velocities

$$c_\mathrm{I} = \sqrt{\frac{\mu}{\rho}}, \quad c_\mathrm{II} = \sqrt{\frac{\lambda + \mu + \nu}{\rho}}, \quad c_\mathrm{III} = \sqrt{\frac{\lambda + 2\mu + \nu}{\rho}}. \tag{101}$$

Note that they satisfy the Pythagorean relation

$$c_\mathrm{I}^2 + c_\mathrm{II}^2 = c_\mathrm{III}^2. \tag{102}$$

This is a universal prediction of our field theory.

## 6 Conclusion

I have described here classical field theories of a ferromagnet and of antiferromagnets with 2 and 3 magnetic sublattices.

## Acknowledgments

These lecture notes reflect the knowledge collected in numerous collaborative projects with students, postdocs, and colleagues. I would like to thank in particular Yaroslaw Bazaliy, David Clarke, Sayak Dasgupta, Se Kwon Kim, Bastian Pradenas, and Oleg Tretiakov.

**Funding information**   The author's research has been supported by the U.S. Department of Energy, Office of Science, Basic Energy Sciences under Award No. DE-SC0019331.

## A   Alternative representations of spin dynamics

The Landau-Lifshitz equation (4) can be recast in other forms. With the aid of vector algebra, we can rewrite it as follows:

$$-\dot{\mathbf{m}} \times S\mathbf{m} - \left(\frac{\partial U}{\partial \mathbf{m}}\right)_\perp = 0 \,. \tag{A.1}$$

Here

$$\mathbf{c}_\perp \equiv \mathbf{m} \times (\mathbf{c} \times \mathbf{m}) = \mathbf{c} - \mathbf{m}(\mathbf{m} \cdot \mathbf{c}) \,, \tag{A.2}$$

is the transverse part of vector $\mathbf{c}$ tangential to the unit sphere $|\mathbf{m}| = 1$. To appreciate the structure of Eq. (A.1), think of $\mathbf{m}$ as the position of a particle constrained to move on a unit sphere. Eq. (A.1) expresses a balance of forces acting on the particle. The second term $-(\partial U/\partial \mathbf{m})_\perp$ is a conservative force generated by potential energy $U(\mathbf{m})$. The first term is a Lorentz force acting on the particle with a unit electric charge moving with velocity $\dot{\mathbf{m}}$ in the magnetic field $\mathbf{b} = -S\mathbf{m}$ of a magnetic monopole of strength $S$ located at the center of the sphere. We can see the spin dynamics expressed by Eq. (A.1) conserves the potential energy:

$$\dot{U} = \dot{\mathbf{m}} \cdot \frac{\partial U}{\partial \mathbf{m}} = \dot{\mathbf{m}} \cdot \left(\frac{\partial U}{\partial \mathbf{m}}\right)_\perp = -\dot{\mathbf{m}} \cdot (\dot{\mathbf{m}} \times S\mathbf{m}) = -S\mathbf{m} \cdot (\dot{\mathbf{m}} \times \dot{\mathbf{m}}) = 0 \,. \tag{A.3}$$

Here we relied on the transverse character of the spin velocity $\dot{\mathbf{m}}$.

Although the spin vector $\mathbf{S}$ has three components, they are not independent in view of the constraint on the length, $|\mathbf{S}| = S$. The same applies to the unit vector $\mathbf{m}$. The most economical description of these vectors would use just two coordinates $\{q^1, q^2\}$ such as the polar angle $\theta$ and azimuthal angle $\phi$:

$$\mathbf{m} = (\sin\theta \cos\phi, \sin\theta \sin\phi, \cos\theta) \,. \tag{A.4}$$

Equations of motion for the new coordinates $\{q^1, q^2\}$ can be derived from the Landau–Lifshitz equation (4) via Eq. (A.1) [16]. They read

$$G_{ij} \dot{q}^j - \frac{\partial U}{\partial q^i} = 0 \,. \tag{A.5}$$

Here indices $i$ and $j$ take on the values of 1 and 2; summation is implied over an index repeated twice—once as a subscript and once as a superscript. The antisymmetric tensor $G$ has the following components:

$$G_{ij} = -S\,\mathbf{m} \cdot \left(\frac{\partial \mathbf{m}}{\partial q^i} \times \frac{\partial \mathbf{m}}{\partial q^j}\right) = -G_{ji} \,. \tag{A.6}$$

Eq. (A.5) expresses a balance of generalized forces acting on coordinate $q^i$. The first term is a gyroscopic force proportional to the generalized velocity $\dot{q}^j$; the second term is a conservative force.

To give an example, we derive the equations of motion for the polar angle $\theta \equiv q^1$ and azimuthal angle $\phi \equiv q^2$. The nonvanishing components of the gyroscopic tensor are

$$G_{\theta\phi} = -S\sin\theta = -G_{\phi\theta}\,. \tag{A.7}$$

We thus obtain

$$\begin{aligned}
-S\sin\theta\,\dot{\phi} - \partial U/\partial\theta &= 0\,, \\
S\sin\theta\,\dot{\theta} - \partial U/\partial\phi &= 0\,.
\end{aligned} \tag{A.8}$$

For a spin in a magnetic field $\mathbf{B} = (0,0,B)$, the Zeeman energy is $U = -\gamma\mathbf{S}\cdot\mathbf{B} = -\gamma SB\cos\theta$. The spin precesses around the direction of the magnetic field at a constant polar angle $\theta$, reflecting the conservation of the spin component $S_z$. The azimuthal angle advances at the rate equal to the Larmor frequency, $\dot{\phi} = -\gamma B$.

It is worth noting that Eq. (A.5) was first derived by Lagrange in a completely different context. He was interested in the long-term evolution of planetary orbits in the presence of a weak perturbation, exemplified by the precession of Mercury's perihelion due to the motion of the Sun perturbed by other planets (mostly Jupiter and Saturn). In that context, $\{q^1, q^2, \ldots, q^6\}$ are parameters of the orbit and $G_{ij}$ is the Lagrange bracket, which is the inverse of the Poisson bracket. See Ref. [17] for details.

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
