# Peer review of "Field theory of collinear and noncollinear magnetic order"

_SciPost Physics Lecture Notes, doi:SciPost Phys. Lect. Notes 85 (2024)_

## Round 1 · Referee Report · Anonymous (Referee 1) · 2024-3-4

Report
The lecture notes give the basics of the theory needed to study ferromagnets and antiferromagnets. The audience should be beginning doctoral students in this field, or it could also serve as a basis for part of a specialized graduate course. Since there is quite a significant number of students being trained in this area, it is very useful to have such lecture notes available.
The flow is smooth and reasonable steps are followed in going from one section (or subsection) to the next. The general approach is to start with a discrete lattice and pass later to a continuum description. This is reasonable and also a helpful point of view for students. Only a few examples of solutions of the equations are given, such as domain walls. This is sufficient given that these lecture notes are kept to a basic level. The most advanced part is in Sec. 5, referring to a 3-sublattice antiferromagnet. The theory for this system is not easily accessible in other texts.
Remarks are following. 1. Eq. (48) gives m in terms of n, for antiferromagnets. This indicates that m=0 for all non-time dependent configurations, e.g., for domain walls. This contradicts various works. For example, in the recent paper H.T. Hirose, et al, Sci. Rep. 7:42440, 2016 (DOI: 10.1038/srep42440), significant net magnetization is observed for antiferromagnetic domain walls. References in this paper indicate that this phenomenon was clear already in previous experimental and theoretical works using models identical to the one explained in the present lecture notes.
-
p 16. The phrase "only use the paramagnetic energy in Eq. (66) in the Landau–Lifshitz equations and leaving out the gradient terms:" is not clear. It would need some reformulation and probably should become more explicit.
-
Eq. (50) and (81) are called in the text "general Landau-Lifshitz equation". The right hand side has the form of a Landau-Lifshitz equation but the left hand side does not. The explanation following Eqs. (81,82) gives a justification for the terminology. But, the behavior of Eq. (81) could be very different than the behavior (solutions, etc) given by the usual Landau-Lifshitz equation. If the author would like to call this a Landau-Lifshitz equation, there would need to be some more justification or explanation or clarification about the terminology.
-
After Eq. (13), a complex amplitude psi is used and the magnetization component mz is set to zero. It should be explained why it is consistent to set mz=0. Note also that, in the beginning of Sec. 3.2.1, it is assumed mz /= 0 for a similar calculation.
Here are some minor or technical remarks. 5. After Eq. (29), it is stated (correctly) that "they can be separated into 4 topological sectors". But, there is no previous explanation about topological sectors. 6. After Eq. (45), the expression "The ellipsis" is unclear. 7. Fields in Eq. (65) have been defined also in Eq. (62,63), although only for a 3-spin system. If possible, it would be better to avoid this repetition (or connect clearly the two definitions).
Typos: "with broken inversion", "a similar permutations", "tou", "sublattce"
I recommend these lecture notes for publication in SciPost as they are well-written and they will be useful to an audience of graduate students.

---

## Round 2 · Referee Report · Anonymous (Referee 1) · 2024-8-20

Report

The author has provided clarifications and made additions that address the remarks of the first report.
I recommend publication of the lecture notes.

Recommendation

Publish (easily meets expectations and criteria for this Journal; among top 50%)

---

## Round 2 · Author Response

I thank the referee for a thorough reading of the manuscript and thoughtful comments. The lecture notes have been expanded to provide better explanations. Specific queries of the referee are addressed below.

---

## Round 2 · List of Changes

1. Indeed, the antiferromagnetic chain has a topological contribution to the total spin that is not included when the uniform magnetization is integrated over the chain length. I have added a new subsection 4.5 Net spin of the antiferromagnetic chain to explain this effect and cited the seminal theoretical works of Faddeev and Takhtajan (for S=1/2) and of Papanicolaou (for classical spins).

  2. The passage has been rewritten to provide a more explicit reference to the included term.

  3. This seems a bit controversial! I have changed the offending terminology and now refer to the "analogs" of the Landau-Lifshitz equation for antiferromagnets or to simply "equations of motion."

  4. An infinitesimal increment of a unit-vector field is orthogonal to the original value (so that the unit length is preserved), hence zero z component. In Sec. 3.2.1, I initially referred to a finite deviation; the subsequent Taylor expansion has no first-order term for the z-component.

  5. Two sentences have been added to explain the meaning of topological sectors.

  6. The term "elipsis" has been replaced with "omitted terms."

  7. The text has been edited to explicitly state that Eq. (80) for the magnetization fields is the same as Eqs. (77) and (78) for three spins. (These are the updated equations numbers reflecting the addition of a new subsection.)

---

## Editorial Decision

published